# Quantitative Assessment of the Relative Impacts of Land Use and Climate Change on the Key Ecosystem Services in the Hengduan Mountain Region, China

**Erfu Dai [1,2], Le Yin [2,3,*], Yahui Wang [1,2], Liang Ma [1,2] and Miao Tong [1,2]**

[1]  Lhasa Plateau Ecosystem Research Station, Key Laboratory of Ecosystem Network Observation and Modeling, Institute of Geographic Sciences and Natural Resources Research, Chinese Academy of Sciences, Beijing 100101, China; daief@igsnrr.ac.cn (E.D.); wangyah.15b@igsnrr.ac.cn (Y.W.); maliang.16b@igsnrr.ac.cn (L.M.); tongm.17b@igsnrr.ac.cn (M.T.)

[2]  University of Chinese Academy of Sciences, Beijing 100049, China

[3]  Key Laboratory of Land Surface Pattern and Simulation, Institute of Geographic Sciences and Natural Resources Research, Chinese Academy of Sciences, Beijing 100101, China

*  Correspondence: yinl.16b@igsnrr.ac.cn

**Abstract:** In the Hengduan Mountain region, soil erosion is the most serious ecological environmental problem. Understanding the impact mechanism of water yield and soil erosion is essential to optimize ecosystem management and improve ecosystem services. This study used the Integrated Valuation of Ecosystem Services and Tradeoffs (InVEST) and Revised Universal Soil Loss Equation (RUSLE) models to separate the relative contributions of land use and climate change to water yield and soil erosion. The results revealed that: (1) Although soil and water conservation has been strengthened in the past 25 years, both water yield and soil erosion increased from 2010 to 2015 due to the conversion of woodland to grassland, which indicates that continuous benefits after the implementation of ecological restoration projects were not obtained; (2) Climate change played a decisive role in water yield and soil erosion changes in the Hengduan Mountain region from 1990 to 2015, and soil erosion was not only related to the amount of precipitation but also closely related to precipitation intensity; (3) The contribution of land use and climate change to water yield was 26.94% and 73.06%, while for soil erosion, the contribution of land use and climate change was 16.23% and 83.77%, respectively.

**Keywords:** water yield; soil erosion; land use change; climate change; relative contribution

---

## 1. Introduction

Ecosystem services are important factors that guarantee human survival and development [1]. Understanding the driving mechanisms of the key ecosystem services is vital for the sustainable management of ecosystems and natural capital [2,3]. Ecosystem services are closely related to topography, land use, soil, biology, climate, and socio-economic factors [4]. Land use and climate change are considered to be the main driving forces of changes in ecosystem services [5,6]. Climate change determines the spatial and temporal distribution of ecosystem services, while land use change affects ecosystem services by changing the ecosystem structure and function [7].

The results of the millennium ecosystem assessment showed that over 60% of the world's ecosystems were degrading [4]. In order to reduce the negative impact of global environmental change, it is essential to strengthen the research on ecosystem service-driven mechanisms. At present, the research methods to determine the ecosystem services' response to impact factors mainly include correlation analysis [8], regression analysis [9], principal component analysis [10], sensitivity analysis [11], geo-detectors [12], and spatial panel models [13]. However, these methods can only explore the impact

of a single factor on ecosystem services. Few studies have focused on separating the relative impacts of land use and climate change. Comparing the simulation results under different conditions provides an effective method to distinguish the impact factors on ecosystem services [14–16]. Ecosystem service assessment models such as Integrated Valuation of Ecosystem Services and Tradeoffs (InVEST) [6], Revised Wind Erosion Equation (RWEQ) [17], and Soil and Water Assessment Tool (SWAT) [18] provide effective methods for calculating the relative contribution of land use and climate change [19–21]. In addition, compared to urban ecosystems, researchers have paid less attention to mountain ecosystem.

The Hengduan Mountain region is an important ecological function area in China. The runoff and hydraulic resources account for 10.9% and 24% of the total water resources in China, respectively [22]. However, due to intense anthropogenic disturbance (vegetation destruction, overgrazing, and sloping land reclamation), the ecological environment has been severely damaged. Annual soil erosion in the Hengduan Mountain region has reached $8.9 \times 10^8$ t, and it is one of the areas that has experienced the most severe soil erosion in China [23]. A series of ecological restoration projects were carried out in the Hengduan Mountain region, aiming to strengthen the capacity of soil and water conservation [24,25]. Therefore, water yield and soil erosion were selected to explore the driving mechanisms of ecosystem services in the Hengduan Mountain region.

The objectives of this study are to: (1) Identify and assess the key ecosystem services from 1990 to 2015 and analyze their spatio-temporal variation characteristics; (2) Separate the relative impacts of land use and climate change on water yield and soil erosion; (3) Propose improvement measures for soil and water conservation in the Hengduan Mountain region.

## 2. Materials and Methods

### 2.1. Study Area

The Hengduan Mountain region (24°29′–33°43′ N, 97°10′–104°25′ E) is located in the southwest of China and is named for its complex terrain [26]. The total area of the Hengduan Mountain region is approximately 450,000 km², and the elevation is 302 m to 7143 m. The regional temperature varies greatly, with an annual mean temperature of 5 °C to 13 °C. The annual total precipitation is 500 mm to 1000 mm, with 60%–90% of the precipitation concentrated from May to October [27]. The main land use types are woodland, grassland, and cropland, accounting for 46.00%, 41.56%, and 7.52% of the total area (2015), respectively (Figure 1).

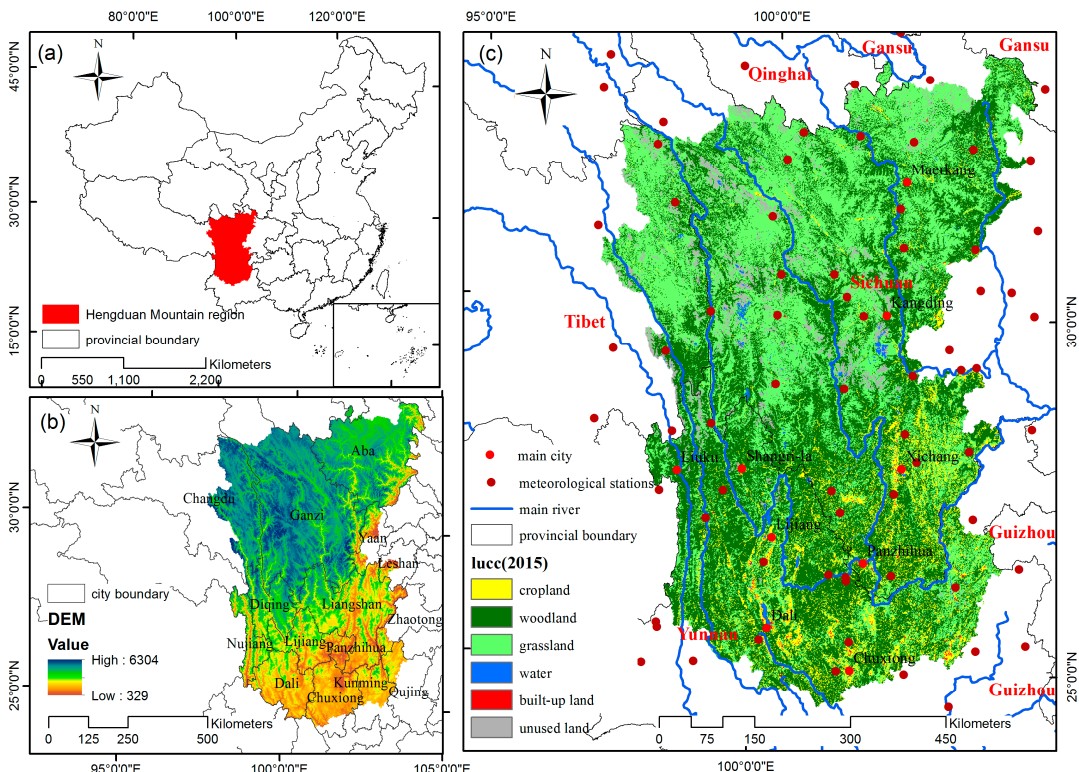

**Figure 1.** (**a**) Location, (**b**) Digital Elevation Model (DEM), and (**c**) land use types in the study area.

## 2.2. Data Preparation

### 2.2.1. Land Use Data

The land use data for the Hengduan Mountain region in 1990, 2000, 2010, and 2015 were derived from the Resource and Environment Data Cloud Platform. These datasets were based on Landsat TM/ETM remote sensing images and generated by manual visual interpretation [28]. All land use data in this study were converted to 1km resolution.

### 2.2.2. Climate Data

Climate data during the period 1990–2015 were derived from the National Meteorological Information Center. This study collected data from 75 meteorological stations in or around the study area and interpolated them to 1km resolution using the ANUSPLIN 42 software developed by the Australian National University to generate the raster data required for the InVEST model [29]. Based on the interpolated climate data, the potential evapotranspiration ($ET_0$) and rainfall erosivity for water yield and soil erosion were calculated. To better represent the average climate condition of the four study periods and avoid the randomly abnormal climate of one specific year, the climate conditions of 1990, 2000, 2010, and 2015 was replaced by the average condition in their adjacent years. For example, the annual average temperature and annual total precipitation of the Hengduan Mountain region from 1998 to 2002 were taken as the annual average temperature and annual total precipitation in 2000.

### 2.2.3. Soil Data

The soil data was obtained from the Harmonized World Soil Database (HWSD), and the plant available water content and soil erodibility were calculated based on the soil texture properties.

### 2.2.4. DEM Data

Given that soil erosion in mountainous areas is sensitive to topographic factors, 90 m resolution DEM data were selected to calculate the slope and length factors in this study. The sub-watersheds were also divided by the DEM data based on the ArcGIS 10.2 platform.

### 2.3. Detecting the Relative Impacts Of Land Use and Climate Change on Ecosystem Services

This study proposes a framework for quantitative evaluation of the impact factors of ecosystem services (Figure 2). Based on remote sensing, meteorology, soil, and DEM data, we first assessed the water yield and soil erosion and analyzed their temporal and spatial dynamics. Then, we separated the relative impacts of land use and climate change on water yield and soil erosion by setting different conditions. Finally, based on the research results, we proposed management strategies to improve regional soil and water conservation capacity in the Hengduan Mountain region.

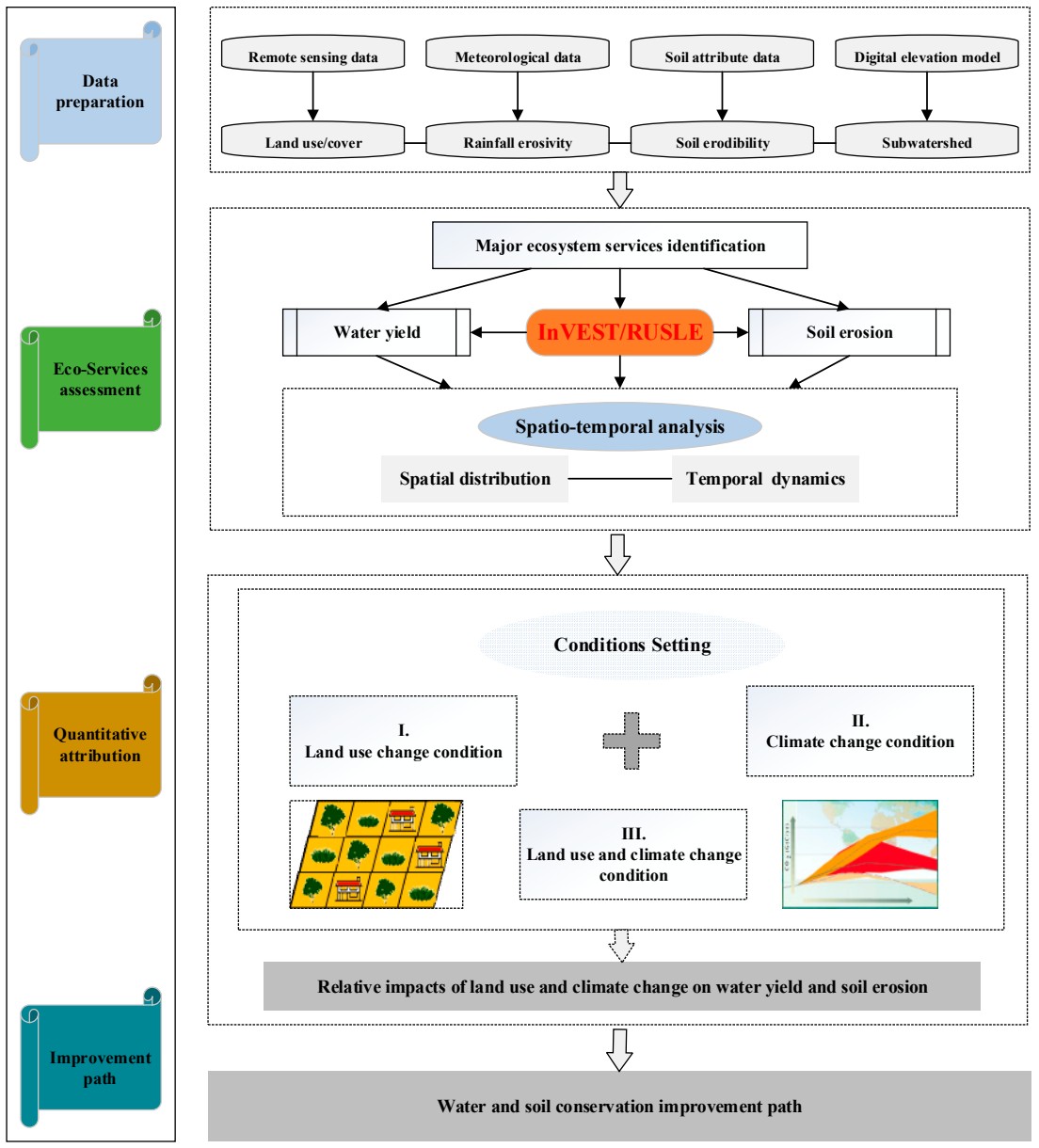

**Figure 2.** Framework of methodology.

## 2.4. Ecosystem Service Assessment

The InVEST model can not only quantify ecosystem services, but also spatialize the assessment results. Currently, the model can be used to evaluate 18 ecosystem services in freshwater, marine, and terrestrial ecosystems [30]. Because this study focuses on the inter-annual variation of ecosystem services, and the data required by the water yield module are easy to obtain, the InVEST model has been selected to assess the water yield in the Hengduan Mountain region. The Revised Universal Soil Loss Equation (RUSLE) model is applicable to soil erosion simulation at different scales and compensates for the limitations of large-scale field observation [31]. Therefore, this study selected the InVEST model and the RUSLE model to assess water yield and soil erosion.

### 2.4.1. Water Yield

The InVEST model simulates water yield on a pixel scale by calculating the difference between precipitation and actual evaporation [30]. The formula is as follows:

$$Y(x) = \left(1 - \frac{AET(x)}{P(x)}\right) \cdot P(x) \tag{1}$$

$$\frac{AET(x)}{P(x)} = 1 + \frac{PET(x)}{P(x)} - \left[1 + \left(\frac{PET(x)}{P(x)}\right)^{\omega}\right]^{\frac{1}{\omega}} \tag{2}$$

$$\omega(x) = Z \cdot \frac{AWC(x)}{P(x)} \tag{3}$$

$$R(x) = k(x) \cdot \frac{ET_0}{P_x} \tag{4}$$

where $Y(x)$ denotes the annual water yield (mm); $AET(x)$ denotes the annual mean actual evapotranspiration (mm); $P(x)$ denotes the annual mean precipitation (mm); $R(x)$ denotes the Budyko aridity index; $ET_0$ denotes the potential evapotranspiration (mm); $AWC(x)$ denotes the plant available water content (mm), determined by soil depth and physicochemical properties; $\omega(x)$ denotes an empirical parameter; $k(x)$ denotes the vegetation evapotranspiration; and $Z$ denotes a seasonal rainfall factor.

AWC was calculated from the soil texture [32]; $k(x)$ was obtained from the irrigation and horticulture handbooks of Food and Agriculture Organization of the United Nations (FAO); the $ET_0$ was calculated by the modified Hargreaves equation [33,34]; and $Z$ used a default value.

### 2.4.2. Soil Erosion

We calculated the soil erosion rates for each grid using the RUSLE model. The formula is as follows:

$$SE = R \times K \times L \times S \times C \times P \tag{5}$$

where SE denotes the annual soil erosion rate (t ha$^{-1}$ yr$^{-1}$); R denotes the rainfall erosivity (MJ mm ha$^{-1}$ h$^{-1}$ yr$^{-1}$); K denotes the soil erodibility (t ha h MJ$^{-1}$ mm$^{-1}$ ha$^{-1}$); LS denotes the slope length–steepness; C denotes the cover management; and P denotes the erosion control practice.

The rainfall erosivity factor R was calculated as follows [35,36]:

$$R = \sum_{i=1}^{12} 1.735 \times 10^{\left(1.5 \log_{10}\left(\frac{P_i^2}{P}\right) - 0.8188\right)} \tag{6}$$

where $P_i$ denotes the monthly rainfall (mm) and P denotes the annual rainfall (mm).

The approach to calculate soil erosivity factor K is as follows [37]:

$$K = \left\{ 0.2 + 0.3e^{[-0.0256SAN(1-\frac{SIL}{100})]} \times \left( \frac{SIL}{CLA+SIL} \right)^{0.3} \times \left[ 1 - \frac{0.25C}{C+e^{3.72-2.95C}} \right] \right.$$
$$\left. \times \left[ 1 - \frac{0.7SN}{SN+e^{-5.51+22.9SN}} \right] \right\} \times 0.1317 \tag{7}$$

where K denotes the soil erodibility factor (t ha h ha$^{-1}$ MJ$^{-1}$ mm$^{-1}$), SAN denotes the sand fraction (%), SIL denotes the silt fraction (%), CLA denotes the clay fraction (%), and C denotes the organic carbon fraction (%). SN = (100 − SAN)/100; the value 0.1317 is the unit conversion coefficient.

The LS was calculated as follows [38,39]:

$$S = 10.8 \times \sin\theta + 0.03(\theta\langle 9\%,\rangle\lambda > 4.6 \text{ m}) \tag{8}$$

$$S = 10.8 \times \sin\theta - 0.50(\theta \geq 9\%, \lambda > 4.6 \text{ m}) \tag{9}$$

$$S = 3.0 \times (\sin\theta)^{0.8} + 0.56(\lambda \leq 4.6 \text{ m}) \tag{10}$$

$$L = \left( \frac{\lambda}{22.13} \right)^{\alpha} \tag{11}$$

$$\alpha = \left( \frac{\beta}{\beta+1} \right) \tag{12}$$

$$\beta = \frac{\sin\theta}{3 \times (\sin\theta)^{0.8} + 0.56} \tag{13}$$

where $\theta$ denotes the slope, $\lambda$ denotes the slope length, $\alpha$ denotes the length-slope exponent, and $\beta$ denotes a factor relative to slope gradient.

C is the ratio of soil loss under a particular crop or vegetation cover to the soil loss from the fallow land. P is the ratio of soil loss with soil and water conservation measures to the soil loss with slope tillage. Factors C and P are available through the RUSLE handbook, published by the United States Department of Agriculture [40]. This study also referred to existing research results to determine the values of factors C and P [41,42].

*2.5. Conditions Setting*

In this study, we set up three conditions and used the ecological models to separate the relative impacts of land use and climate change on water yield and soil erosion [43] (Table 1). Under the land use and climate change condition, both land use and climate data were input into the model according to the actual situation. Under the land use change condition, the climate was the same as that in 1990, and 2015 land use data were input into the model. Under the climate change condition, the land use was the same as that in 1990, and the climate data of 2015 were input into the model.

**Table 1.** Conditions setting.

| Year | Land Use Change + Climate Change Condition | Land Use Change Condition | Climate Change Condition |
|---|---|---|---|
| 1990 | 1990 land use data 1990 climate data | 1990 land use data 1990 climate data | 1990 land use data 1990 climate data |
| 2015 | 2015 land use data 2015 climate data | 2015 land use data 1990 climate data | 1990 land use data 2015 climate data |

## 3. Results

### 3.1. Driving Forces Analysis of Water Yield and Soil Erosion

#### 3.1.1. Land Use Change Analysis

Affected by the implementation of ecological restoration projects, woodland and grassland were frequently interconverted, while cropland, water bodies, built-up land, and unused land only slightly changed. The increase of woodland was mainly caused by the conversion of grassland, among which the woodland increased by 4560 km$^2$ and the grassland decreased by 4113 km$^2$. Moreover, both woodland and grassland changed from low to high coverage. The cropland and unused land decreased by 768 km$^2$ and 649 km$^2$, respectively , and were mainly converted to grassland and woodland. Cropland was converted to built-up land, which increased by 764 km$^2$. There was little fluctuation in water body areas. The characteristics of land use conversion were different in each period, with the period from 2000 to 2010 being the most dramatic period of land use change (Figure 3).

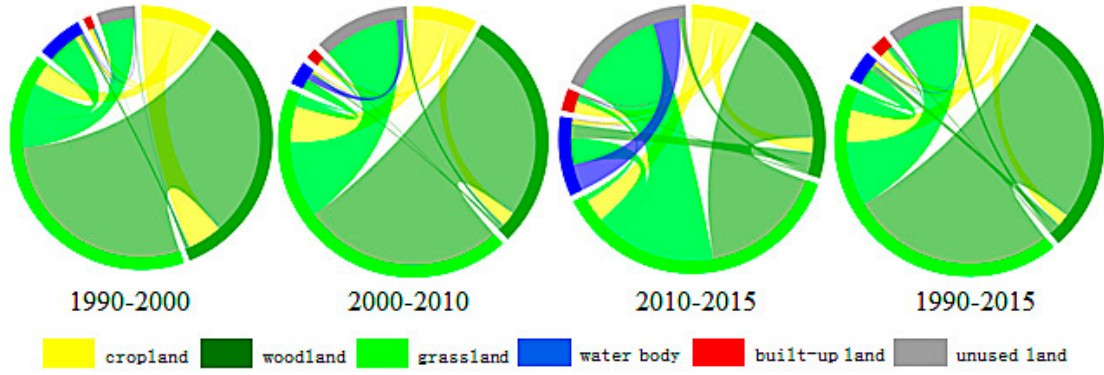

**Figure 3.** Land use change during different periods.

#### 3.1.2. Climate Change Analysis

The change rates of the annual mean temperature and annual total precipitation in the Hengduan Mountain region were 0.05 °C yr$^{-1}$ and −2.56 mm yr$^{-1}$, respectively. This indicated that the Hengduan Mountain region became warmer and dryer over the 25-year period. Climate change also exhibited significant spatial heterogeneity. From 1990 to 2015, the northern area of the Hengduan Mountain region became warmer and wetter, while most of the remaining areas became warmer and dryer (Figures 4 and 5).

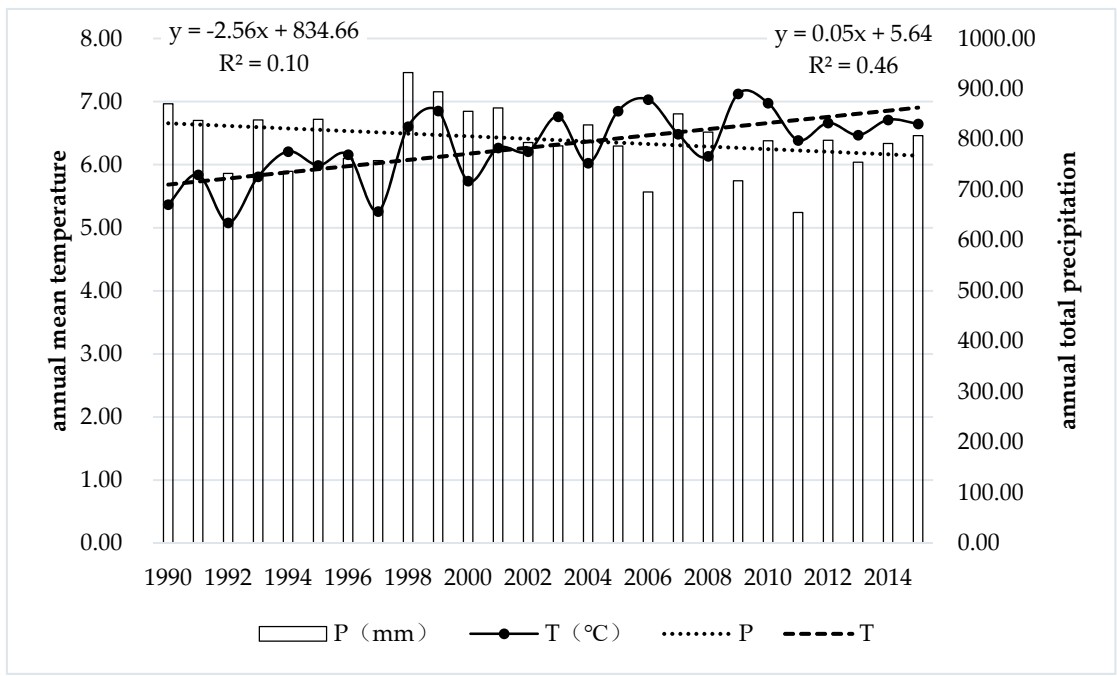

**Figure 4.** Changes in annual mean temperature (T) and annual total precipitation (P) in the Hengduan Mountain region during the period 1990–2015.

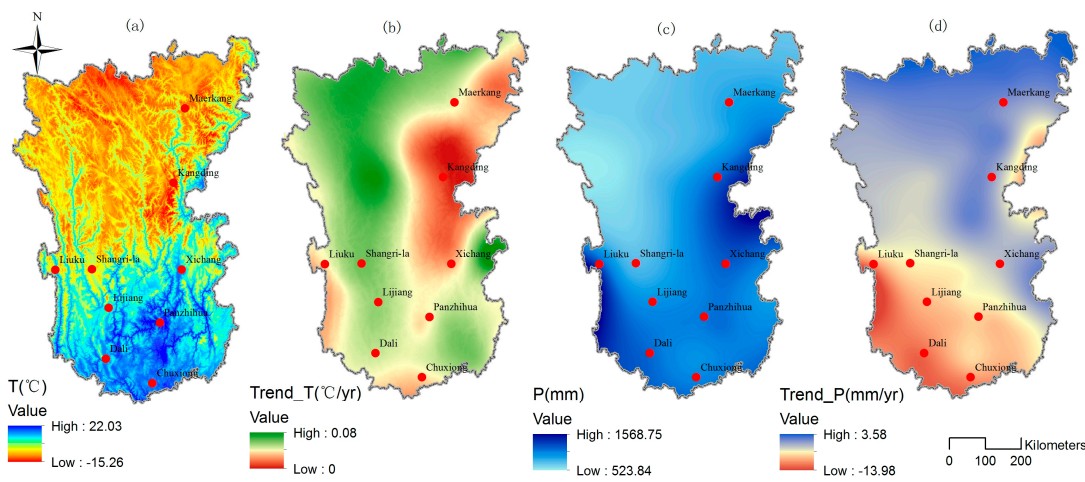

**Figure 5.** Spatial distribution of (**a**) annual mean temperature (T), (**b**) annual mean temperature change trend (Trend_T), (**c**) annual total precipitation (P), and (**d**) annual total precipitation change trend (Trend_P) from 1990 to 2015.

### 3.2. Assessment, Validation, and Spatio-Temporal Heterogeneity of Water Yield and Soil Erosion

The water yield in 1990, 2000, 2010, and 2015 was 412.04 mm, 467.36 mm, 377.76 mm, and 405.62 mm, respectively, while the soil erosion rates were 28.61 t ha$^{-1}$ yr$^{-1}$, 31.23 t ha$^{-1}$ yr$^{-1}$, 24.77 t ha$^{-1}$ yr$^{-1}$, and 25.86 t ha$^{-1}$ yr$^{-1}$ in 1990, 2000, 2010, and 2015, respectively. The simulated water yield in the Hengduan Mountain region was similar to 421.07 mm of the 10-km resolution multi-year average surface runoff datasets. The simulation value of soil erosion was similar to 26.86 t ha$^{-1}$ yr$^{-1}$ of Ge et al. (2014) and 27.31 t ha$^{-1}$ yr$^{-1}$ of Ouyang et al. (2016) [44,45].

In general, the spatial distribution pattern of water yield and soil erosion had not changed from 1990 to 2015. The areas with high water yield were mainly distributed in the southwest and northeast regions. The areas with severe soil erosion (>50 t ha$^{-1}$ yr$^{-1}$) were mainly located in the valleys of the Hengduan Mountain region [46] (Figure 6).

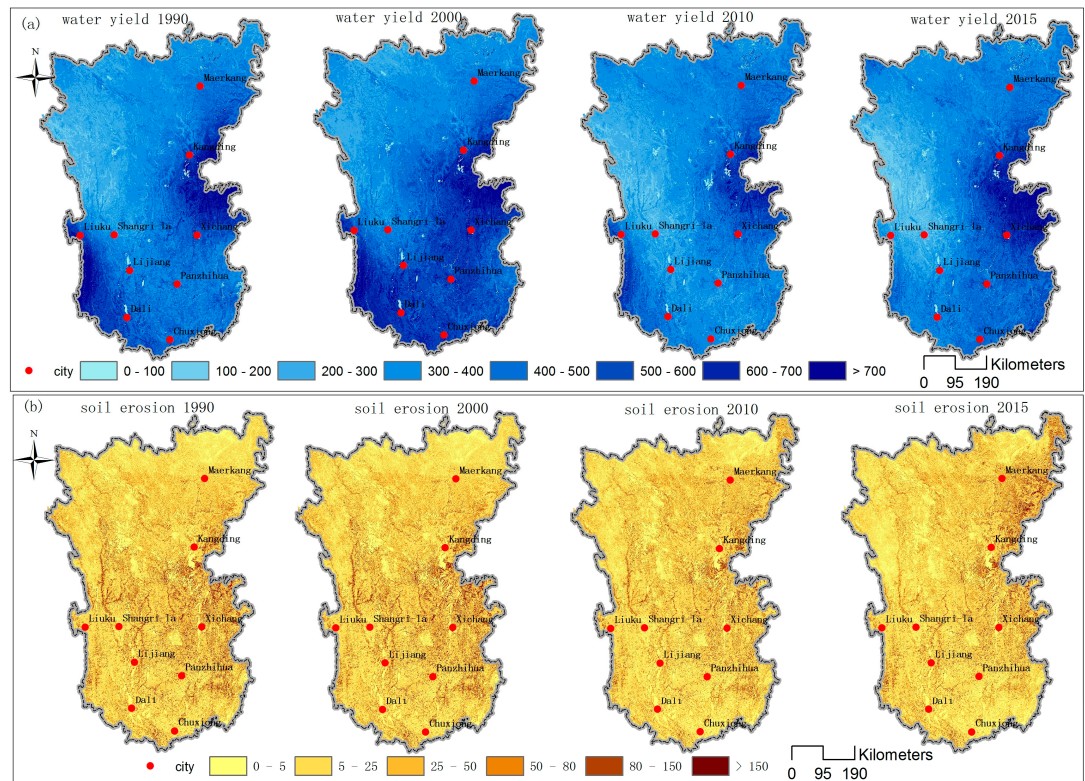

**Figure 6.** Assessment results of (**a**) water yield (mm) and (**b**) soil erosion (t ha$^{-1}$ yr$^{-1}$).

### 3.3. Changes in Water Yield and Soil Erosion under Different Conditions

Under the land use and climate change condition, the water yield in the Hengduan Mountain region decreased by 6.42 mm, and the soil erosion decreased by 2.75 t ha$^{-1}$ yr$^{-1}$ from 1990 to 2015. There was significant spatial heterogeneity in the trends of water yield and soil erosion rates in different areas. The decreased water yield was mainly concentrated in the southwest and northeast regions of the Hengduan Mountain region, while the increased area was mainly distributed in the northwest and southeast. Soil erosion decreased in most areas except in the northeast Hengduan Mountain region. Water yield and soil erosion decreased only during the period 2000–2010 (Figure 7a, Figure 8).

Under the land use change condition, water yield decreased by 1.73 mm, while soil erosion decreased by 0.43 t ha$^{-1}$ yr$^{-1}$ in the Hengduan Mountain region from 1990 to 2015. For different periods, land use change led to a decreasing trend of both water yield and soil erosion from 1990 to 2010. The water yield and soil erosion increased only during the period 2010–2015 (Figure 7b, Figure 9).

From 1990 to 2015, the water yield and soil erosion rates decreased by 4.69 mm and 2.22 t ha$^{-1}$ yr$^{-1}$ under the climate change condition, respectively. The increasing area of water yield was mainly concentrated in the northwest and southeast, while the decreasing area was mainly distributed in the south and southwest Hengduan Mountain region. Soil erosion decreased in most areas, and the increasing area was mainly concentrated in the northeast and parts of the southwest areas. For different periods, water yield and soil erosion decreased only during the period 2000–2010 (Figure 7c, Figure 10).

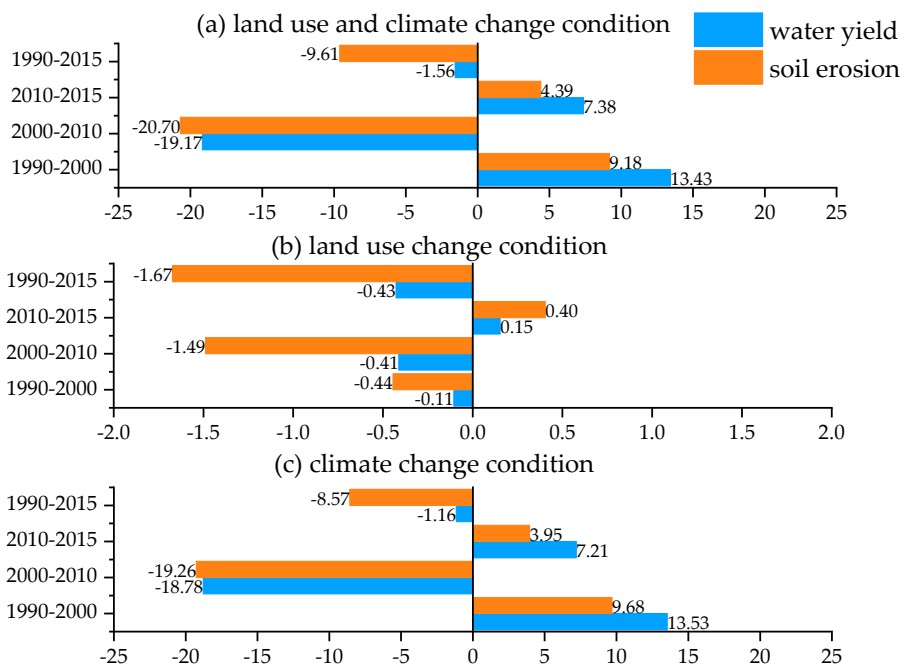

**Figure 7.** Change rates of water yield and soil erosion under different conditions in different periods (%).

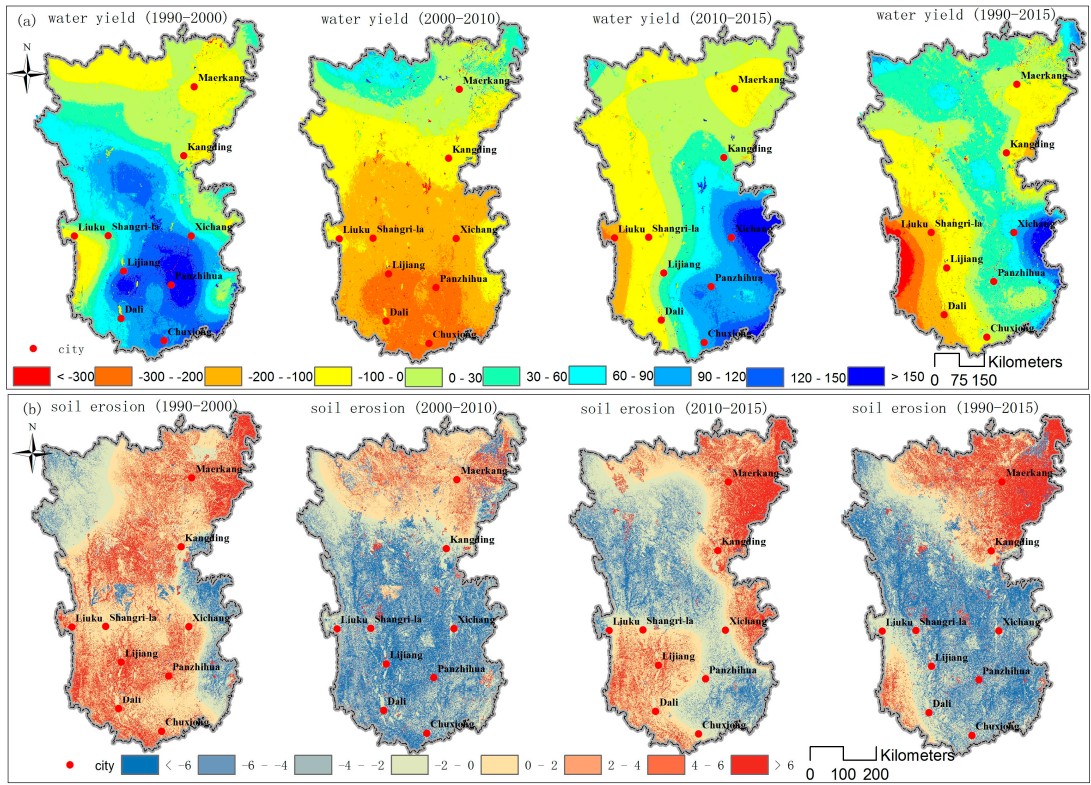

**Figure 8.** Spatial variation in (**a**) water yield (mm) and (**b**) soil erosion (t ha$^{-1}$ yr$^{-1}$) under the land use and climate change condition.

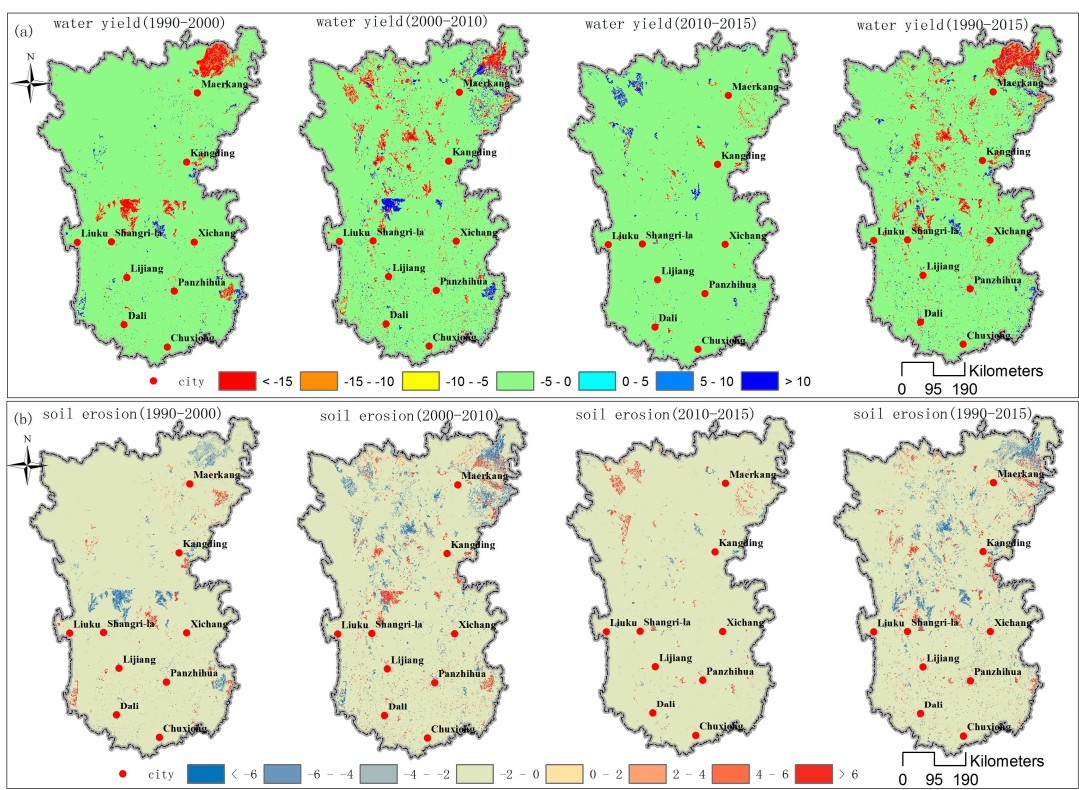

**Figure 9.** Spatial variation in (**a**) water yield (mm) and (**b**) soil erosion (t ha$^{-1}$ yr$^{-1}$) under the land use change condition.

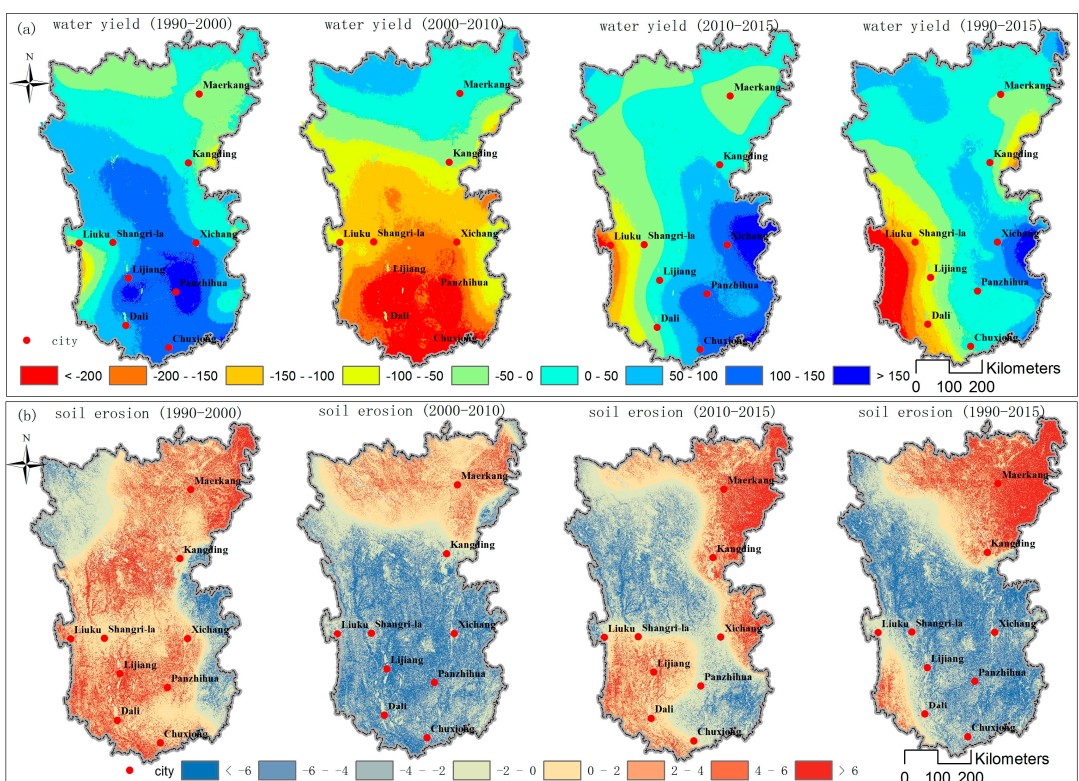

**Figure 10.** Spatial variation in (**a**) water yield (mm) and (**b**) soil erosion (t ha$^{-1}$ yr$^{-1}$) under the climate change condition.

## 4. Discussion

### 4.1. Impacts of Land Use Change on Water Yield and Soil Erosion

From 1990 to 2015, the area of land use conversion was 15761 km$^2$, accounting for 3.52% of the total area. Because the main type of unused land in the Hengduan Mountain region was bare rock, we used bare rock instead of unused land for statistical analysis and did not consider the soil erosion. The current study determined the water and soil conservation capacity according to the changing trend of water yield and soil erosion rate during the land use conversion. The water yield capacity was: bare rock > cropland > grassland > woodland, and the soil conservation capacity was: woodland > grassland > cropland (Table 2, Figure 11).

**Table 2.** Impacts of primary land use conversion on water yield and soil erosion (1990–2015).

| Land Use Change | Area (km$^2$) | Water Yield | | Soil Erosion | |
|---|---|---|---|---|---|
| | | Mean (mm) | Sum ($10^8$ m$^3$) | Mean (t ha$^{-1}$ yr$^{-1}$) | Sum ($10^6$ t) |
| Cropland to woodland | 753 | −181.94 | −1.37 | −103.85 | −7.82 |
| Cropland to grassland | 1017 | −121.93 | −1.24 | −55.75 | −5.67 |
| Woodland to cropland | 706 | 177.05 | 1.25 | 94.9 | 6.70 |
| Woodland to grassland | 3641 | 48.06 | 1.75 | 36.94 | 13.45 |
| Grassland to cropland | 852 | 122.07 | 1.04 | 42.61 | 3.63 |
| Grassland to woodland | 8429 | −54.93 | −4.63 | −30.75 | −25.92 |
| Grassland to bare rock | 1160 | 130.34 | 1.51 | −51.21 | −5.94 |
| Bare rock to grassland | 2196 | −101.09 | −2.22 | 49.18 | 10.80 |

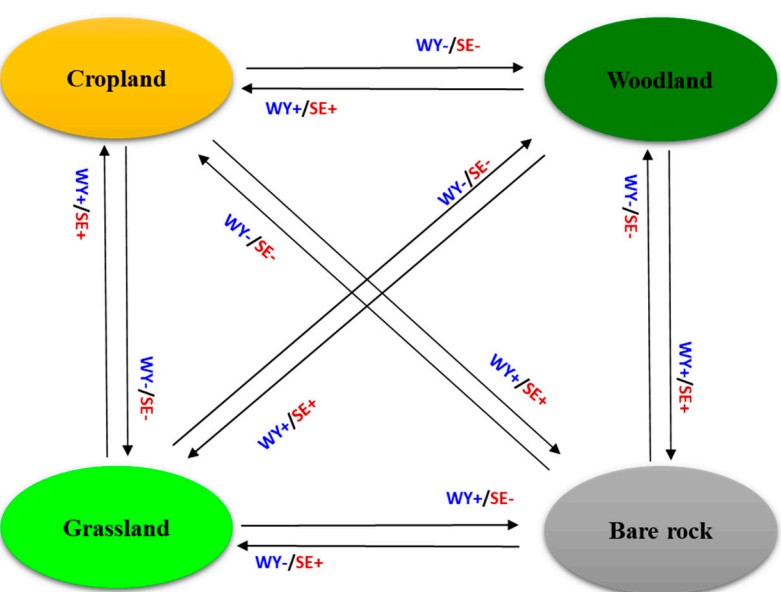

**Figure 11.** Change trends in water yield (WY) and soil erosion (SE) among major land use conversions.

Under the impact of land use change, water yield and soil erosion have decreased over the past 25 years. However, compared with the period 1990–2010, both water yield and soil erosion showed an increasing trend from 2010 to 2015, which was mainly caused by the conversion of woodland to grassland. Given the continuous implementation of ecological restoration projects in this region, this indicated that woodland degraded from 2010 to 2015, which may be related to forest harvesting, pests and diseases, and fire. Studies also found that, from 1998 to 2014, the role of human activities in improving the net primary productivity (NPP) of vegetation significantly decreased [47]. Thus, forest dynamic monitoring, such as that carried out by satellite [48] and drones [49], should be strengthened in the future so that the continuous benefits of ecological restoration projects can be achieved.

The current study found that the implementation of ecological restoration projects in the Hengduan Mountain region can reduce soil erosion, but also inevitably reduce water yield, which poses a certain threat to agricultural irrigation and residential water in downstream areas. In the Loess Plateau region of China, studies have shown that vegetation productivity is already close to the maximum NPP, and continuing revegetation will seriously threaten the water security of human beings [50]. Moreover, influenced by lithology, topography, climate, and human activities, rocky desertification occurred in the Sichuan and Yunnan Provinces in an area of approximately 37,400 km$^2$, accounting for approximately 26.66% of the exposed carbonate rock area and 4.32% of the total area [51]. Although precipitation is abundant in the Hengduan Mountain region, the loss of vegetation and soil would exacerbate infiltration, and the amount of water stored in the soil is low, resulting in insufficient soil moisture for vegetation growth [52]. Thus, future research on ecosystem service trade-offs [53,54] should be strengthened to find a balance between ecological and socio-economic demands to maximize the possibility of achieving the dual goals of ecological protection and economic development. In addition, due to the limitations imposed by topography, croplands are scarce in the Hengduan Mountain region. According to the requirements of the Sichuan, Yunnan, and Tibet Land Use Master Plan (2006–2020), we should strictly protect croplands and strengthen the construction of basic croplands in the future in order to ensure a stable and sustained food supply in this region. Therefore, ecological restoration by returning cropland to forest and grassland is neither permissible nor desirable at present.

*4.2. Impacts of Climate Change on Water Yield and Soil Erosion*

Temperature and precipitation are the most important climatic factors affecting regional ecosystem services [55,56]. From 1990 to 2015, the area that experienced a warming and drying trend was 307,618 km$^2$, accounting for 68.87% of the total area, while the area that underwent a warmer and wetter trend was 139,019 km$^2$, accounting for 31.13% of the total area. Without the impact of land use change, the water yield and soil erosion rates in the warming and drying trend area decreased by 28.05 mm and 5.99 t ha$^{-1}$ yr$^{-1}$, respectively, while in the warming and wetting area, the water yield and soil erosion rates increased by 46.96 mm and 6.12 t ha$^{-1}$ yr$^{-1}$, respectively (Table 3).

**Table 3.** Changes in water yield and soil erosion under the climate change condition from 1990 to 2015.

| Climate Change | Area (km$^2$) | Water Yield (mm) | Soil Erosion Rates (t ha$^{-1}$ yr$^{-1}$) |
|---|---|---|---|
| Warming and drying | 307,618 | −28.05 | −5.99 |
| Warming and wetting | 139,019 | 46.96 | 6.12 |

During each period, the change trend for water yield and soil erosion was consistent with precipitation, which indicated that precipitation was the dominant factor affecting water yield and soil erosion. In addition, although the precipitation decreased from 976.04 mm to 847.45 mm from 1990 to 2015, rainfall erodibility increased from 3164.56 MJ mm ha$^{-1}$ h$^{-1}$ yr$^{-1}$ to 3922.89 MJ mm ha$^{-1}$ h$^{-1}$ yr$^{-1}$. Therefore, soil erosion was not only affected by the amount of precipitation but it may also have been closely related to precipitation intensity, which is consistent with previous research findings [57,58].

Studies have shown that extreme drought, rainfall, and frequent flooding, which all pose a great threat to the ecological environment and the safety of people's lives and property, have significantly increased in the past few decades in most parts of southwest China [59,60]. In response to climate change, the construction of terraces and walls on hill slopes in southern Yunnan Province, such as the famous Hani terraces, provide a reference for the Hengduan Mountain region to prevent further soil erosion and realize sustainable agricultural development [61]. In addition, the eco-hydrological effects of land use change caused by human beings should be reviewed before the implementation of any ecological projects. Ideal vegetation for use in the ecological restoration of the Hengduan Mountain region would possess the characteristics of low water consumption and strong soil and water

conservation capacity. For example, research has found that primary and old growth dark coniferous forests consume less water than other vegetation types in southwest China [62].

### 4.3. Contribution of Land Use and Climate Change to Water Yield and Soil Erosion

By comparing the changes in water yield and soil erosion under different conditions, we calculated the contribution of land use change and climate change to water yield and soil erosion from 1990 to 2015. For water yield, the contribution of land use and climate change was 26.94% and 73.06%, while for soil erosion, the contribution of land use and climate change was 16.23% and 83.77%, respectively. Therefore, climate change was the main driving force affecting the spatial and temporal changes in water yield and soil erosion from 1990 to 2015.

### 4.4. Uncertainties

The premise of separating ecosystem service impact factors is that different factors are independent of each other and have no interactions. But, in fact, there are complex interactions between land use and climate change that jointly drive global environmental change [63,64]. Therefore, the separation method may affect the accuracy of the contribution of land use and climate change to water yield and soil erosion. There are also uncertainties in the ecosystem service assessment. For water yield, the InVEST model does not distinguish between surface water, groundwater, and base flow, and does not consider the interaction between them [30]. The model greatly simplifies the consumption demand, and determining the demand through land use types may not fully reflect the distribution of water resources on different utilization patterns and time scales [8]. For soil erosion, there are few studies on cover management factor, C, and erosion control practice factor, P, in this region; the current study refers to the values of existing studies, which may also lead to uncertainty. This study only verified the assessment results of water yield and soil erosion with the help of existing research results. In the future, field test data or relevant data (such as evapotranspiration data and soil erosion survey data) could be used to verify the assessment results pixel by pixel in order to improve the reliability of the assessment results [65,66].

## 5. Conclusions

This study proposed a framework for the separation of ecosystem service impact factors and quantified the contribution of land use and climate change to water yield and soil erosion in the Hengduan Mountain region. The main conclusions were:

(1) The total amount of water yield and soil erosion decreased under the impact of land use and climate change in the Hengduan Mountain region from 1990 to 2015, which indicates that the soil and water conservation capacity has increased over the past 25 years.

(2) Under the land use change condition, soil and water conservation capacity in the Hengduan Mountain region has decreased in the last five years. Dynamic monitoring of forests should be strengthened in the future so that ecological restoration projects can bring about continuous benefits.

(3) Climate change has played a decisive role in the change of water yield and soil erosion in the Hengduan Mountain region. Precipitation was the main factor affecting water yield and soil erosion, and soil erosion was not only affected by the amount of precipitation, but it was also closely related to precipitation intensity.

(4) The contribution of land use and climate change to water yield was 26.94% and 73.06%, respectively, while for soil erosion, the contribution of land use and climate change was 16.23% and 83.77%, respectively.

**Author Contributions:** E.D. conceived the study. L.Y. analyzed the results and wrote the manuscript. Y.W., L.M., and M.T. assisted with data processing. All authors have read and agreed to the published version of the manuscript.

**Funding:** This research has no external funding.

**Acknowledgments:** This research was funded by the National Key R&D Program of China (Grant No. 2017YFC1502903, 2018YFC1508805), the National Natural Science Foundation of China (Grant No. 41530749), and the Strategic Priority Research Program of Chinese Academy of Sciences (Grant No. XDA19040304).

**Conflicts of Interest:** The authors declare no conflict of interest.

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
