# Peer review of "Quantitative Assessment of the Relative Impacts of Land Use and Climate Change on the Key Ecosystem Services in the Hengduan Mountain Region, China"

_sustainability, doi:10.3390/su12104100_

Round 1

Reviewer 1 Report

I appreciate the great efforts that the authors have made in response to my previous questions and concerns. I still have a few comments on the revised manuscript.

Conclusion #1, “both water yield and soil erosion increased from 2010 to 2015 due to the 21 impact of land-use change, which indicated that long-term benefits after the implementation 22 of ecological restoration projects were not obtained” - This is an interesting argument. Why did the land-use change from 2010 to 2015 “impede” the long-term benefits of ecological restoration projects? If this is real, the authors should check what kind of land-use change would increase water yield and erosion.

Conclusion #3, “The contribution of land use and climate change to water yield was 26 26.94% and 73.06%, while for soil erosion, the contribution of land use and climate change 27 was 16.23% and 83.77%, respectively”. How was the contribution calculated? Which numbers were used in Figure 8, 10, 12?

Conclusion #4, “strengthening forest dynamic monitoring, and selecting crop vegetation with less water consumption” - These might be good strategies for future ecological restoration, but they not involved in the authors’ experiments and results, and not supported in their results. Therefore, this should not be the author’s conclusion.

Author Response

Dear reviewer #1:

Thank you for your valuable comments and suggestions to improve our manuscript submitted to Sustainability. Following these, we have made substantial revisions, with detailed responses to each of your questions and suggestions given below:

Note:In order to find the modified location easily, we have marked the modified location in the reply letter (the final state of the revised manuscript).

---------------------------------------------------------------------------------------------------

Response to specific comments:

(1) From 2010 to 2015, under the land use change condition, the increase of water yield and soil erosion in the Hengduan Mountain region was mainly caused by the decrease of woodland. So, according to your request, we modified “both water yield and soil erosion increased from 2010 to 2015 due to the impact of land-use change” to “both water yield and soil erosion increased from 2010 to 2015 due to the decrease of woodland”.(Line 21)

(2) In section 4.3, we have explained that the contribution of land use and climate change to water yield and soil erosion were calculated by comparing the changes in water yield and soil erosion under land use change condition and climate change condition (Line 333-335). For example, from 1990 to 2015, the water yield under land use condition and climate change condition increased by 2 mm and 4 mm, respectively, so the contribution of land use change and climate change to water yield was 33.33% and 66.67%, respectively.

(3) According to your request, in the abstract and conclusion section of the revised manuscript, we deleted relevant contents of the future ecological restoration strategies.

Thank you again for your valuable comments. We hope the changes we made to the manuscript meet your expectations.

Reviewer 2 Report

Dai et al. quantitatively estimates the relative impacts of land use and climate change on two key ecosystem services: water yield and soil erosion in the Southwest region of China. This research is designed appropriately and the paper is well-organized. The results are clearly represented and they well support the conclusions of this research. However, the methods of this study have not been described adequately and the uncertainties of simulation results should be analysed and discussed more. In specific:

1) There are many models can simulate the ‘water yield’, and many of these models run at fine time step (e.g. daily) and explicitly simulate the surface and underground soil water. So why this study select the InVEST model, which runs at yearly time step and does not distinguish surface and underground water. An explanation of model selection should be provided in the method section.

2) Impacts of land cover change on soil erosion are mainly reflected though the cover management factor (C) and erosion control practice factor (P). However, the method for calculating C and P is only described as ‘The values of C and P were obtained from the previous research’. More detailed description is necessary.

3) Uncertainties of simulation results should be analysed and discussed more. The description that the simulated water yield and soil erosion rate in this study is ‘very similar to previous estimations’ is quite vague. More quantitative assessment of the simulation results in this study should be conducted. Some databases of evapotranspiration have been produced based on satellite data (e.g. MODIS-ET, GLASS-ET) or data driving method (e.g. Jung et al., 2010, Nature). All these databases can be used to assess the uncertainties in ET and water yield in this study. China has conducted twice national soil erosion surveys (Yue et al., 2016, PNAS). Uncertainties in the simulated soil erosion rate of this study can be assessed based on the results of national erosion surveys.

In addition, the English of this paper should be improved by a native English speaker.

Minor comments:

Line 20-23. The conclusion ‘which indicated that long-term benefits after the implementation 22 of ecological restoration projects were not obtained’ is incorrect. First, the increase of water yield due to ecological restoration projects is indeed one of the ‘long-term benefits’. Second, although these projects did not stop the increase of soil erosion, they still slowed down the increase in soil erosion rate. I suggest to delete this conclusion.  Two commas in ‘impact of land use change,, which indicated that long-term’.

Line 103-108. What is the definition of ‘extreme years’? To better represent the average climate condition of the four study periods and avoid the randomly abnormal climate of one specific year, the climate condition of 1990, 2000, 2010 and 2015 (all study periods) should all be replaced by the average condition in their adjacent years.

Line 133. The full name of InVEST should be given when the it first appears in this paper (i.e. line 53).

Line 190. The simulated impacts of land use change on soil erosion are depend strongly on the accuracy of C and P factor. Thus, the method for calculating C and P should be explained detailedly. For example, the equations used to calculate the C and P factor, or the lookup table of C and P factor for different vegetation types (at least in the supporting information)

Line 206 analysis

Fig.4 Two decimal places are enough for the coefficients of the regression functions, and the R2. The abbreviations P and T should be explained in the figure caption

Fig.5 All abbreviations (T, Trend_T, P, Trend_P) should be explained in the figure caption (also for other figures, e.g. WY, SE in Figs 6, 7, 9, 11). Units of Trend_T and Trend_P should be oC yr^-1 and mm yr^-1 respectively.

To decrease the number of figures, figures 8, 10, 12 can be combined into one figure (with 3 sub-plots).

Lines 239-244. A quantitative comparison between simulation results in this study to previous studies is necessary. The ‘very similar’ is a very vague description.

Author Response

Dear reviewer #2:

Thank you very much for your kind comments and helpful suggestions for our manuscript submitted to Sustainability. We have made substantially revisions according to your comments, for which our responses are found below:

Note:In order to find the modified location easily, we have marked the modified location in the reply letter (the final state of the revised manuscript).

Response to comments:

Major comments

(1) This study mainly focuses on the inter-annual variation of ecosystem services, and the data required by the InVEST model are easy to obtain. Therefore, the InVEST model was selected to assess water yield in the Hengduan Mountain region. We added the reason for selecting the InVEST model in the revised manuscript. (Line 121-124)

(2) According to your request, we added the definition of C and P factors and their data sources in the revised manuscript. The specific contents are as follows: C is defined as the ratio of soil loss under a particular crop or vegetation cover to the corresponding loss from the fallow land. P is defined as the ratio of soil loss with soil and water conservation measures to the corresponding soil loss with up and down slope tillage. Factors C and P are available through the RUSLE handbook published by the United States Department of Agriculture. (Line 169-173)

(3) As you said, there is some uncertainties in verifying the evaluation results only through existing studies. Relevant data should be used to verify the evaluation results pixel by pixel in order to make the apple results more precise. In the uncertainties section of the revised manuscript, we added a description of how to verify the evaluation results in the future. (Line 351-355)

(4) We have polished the language of this manuscript through Charles worth.

Specific Comments:

(1) First of all, the decrease of water yield and soil erosion indicates the enhancement of soil and water conservation capacity in the Hengduan Mountain region, while the increase of water yield and soil erosion indicates the weakening of soil and water conservation capacity. The ecological restoration project did not bring long-term benefits, which meant that under the land use change condition, the water yield and soil erosion showed an increasing trend in the last 5 years compared with the previous 20 years (Figure 10). In order to avoid misunderstanding, we changed the long-term benefits to the continuous benefits. (Line21 , Line276 , Line 365-366)

(2) We are very sorry that our expression has caused your misunderstanding. We have adopted your suggestion and added the correct expression in the revised manuscript. The content added is as follows: To better represent the average climate condition of the four study periods and avoid the randomly abnormal climate of one specific year, the climate condition of 1990, 2000, 2010, and 2015 was replaced by the average condition in their adjacent years. (Line 93-95)

(3) We added the full name of InVEST model in the revised manuscript. (Line 49)

(4) See major comments 2.

(5) In the revised manuscript, we modified analyses to analysis. (Line 184)

(6) We reduced the regression coefficient and R2 in Figure 4 to two decimal places, and explained the P and T in the figure caption. (Line 205)

(7) We explained all the abbreviations in the figure caption and modified the units. (Line 206-207, Line 208-211 , Line 224, Line 247, Line 250, Line 253)

(8) Figure 8,10,12 in the original manuscript were merged in the revised manuscript. (Line 256)

(9) We have supplemented the evaluation results in the existing studies of  the revised manuscript to make the verification more clear. (Line 213-219)

This manuscript is a resubmission of an earlier submission. The following is a list of the peer review reports and author responses from that submission.

Round 1

Reviewer 1 Report

The manuscript presents a quantitative attribution framework for ecosystem services, and used the Hengduan Mountain region as a study area to separate the relative contribution of land use and climate change to water yield and soil erosion. The topic is interesting and important. In general, the work was designed well, the presentation of results and discussion were well organized. It seems to be suitable for publication in Sustainability. However, they are few points need to be clarified before it can be accepted.

  • Novelty of the study, the authors pointed out their objectives clearly, but the novelty of the study is missing, which need to be stressed in the revised version.
  • Basic methodology is missing in the abstract.
  • Conclusion need to be rewritten, more concise and direct conclusion is appreciated, avoid citation in the conclusion section. Normally, conclusion only summarize the whole work and point out the future research.
  • The use of English language needs check by native speaker.

Reviewer 2 Report

The study by Dai evaluates the impacts of land use and climate change on water yield and soil erosion in the Hengduan Mountain region. The authors found that climate change plays a more important role in affecting the water yield and soil erosion, and also provided some strategies to improve soil and water conservation capacity. This study is on a topic of relevance and general interest to the readers of this journal. However, there are quite a few flaws in the methodology and conclusions, that need some improvement and further discussion.

My concerns are:

  1. The experimental design of separating land use and climate change effects is not clear in Section 2.5. How long is each simulation? Are they continuous simulations from 1990 to 2005? For the land-use change condition (without climate change), which year of climate data was used? For the climate change condition (without land use), which year of land cover condition was used in the simulation. 
  1. Because the experimental design is not clearly described, the conclusion about the dominant effects of climate change is very questionable. According to Figure 11 and 12, I assume the identified impacts of climate change on water yield and soil erosion is simply based on the difference between individual years. For instance, the year 2010 is drier than the year 2000 (Figure 4), so we see decreased water yield and soil erosion. In other words, the identified impacts are mainly from the inter-annual variability of temperature and precipitation, but not the long-term climate change effects. A better way to quantify the climate change impacts is to conduct continuous simulations from 1990 to 2015 with a constant land cover, the trend (or the difference between two certain periods, such as 2006-2015 minus 1990-2000) can be considered as impacts of climate change. 
  1. A few conclusions are just the authors’ speculations, and not actually supported by their results. The authors mentioned that “soil erosion was not only related to the amount of precipitation but also closely related to precipitation intensity”. However, this study never examined the effects of precipitation intensity.

  1. The authors imply that “the implementation of ecological restoration projects in the Hengduan Mountain region had not brought long-term benefits” because land-use impacts play a minimal role in water yield and soil erosion. However, in the discussion on future soil and water conservation, the authors never mentioned the impacts of climate change, but only proposed a few land-use related strategies. Will these strategies really work, or not bring “long-term benefits” as the historical “ecological restoration projects”? 

Specific Comments:

Abstract:

The first two sentences do not have any connections.

Ecosystem service is mentioned many times throughout the manuscript. What is the relationship between the Ecosystem service and soil erosion (or water yield)? 

Highlight #5,  how about the implications of future climate change?

Data:

Provide spatial resolutions of all the datasets. How to deal with the data with different resolutions.

What is "extreme temperature"? Daily maximum/minimum?

Provide a map of the meteorological stations.

Figure 2. What is “improvement approach”? How is it reflected in the results? What is the difference between the two “Relative impacts of land use and climate change on water yield and soil erosion” ?

Section 2.4: 

What is the relationship between InVEST and RUSLE? I couldn't find RUSLE in Figure 2.

How to identify Kxj (vegetation evapotranspiration coefficient)?

What is PAWC?

Which variable in the soil loss equation considers the land cover type, and how to identify its value for each land cover type?

Section 2.5:

See the major comment #1.

Figure 4:

The bar for precipitation in 1990 is covered by the text box.

Are the values averaged over the 75 stations or all the grid cells in Figure 5?

Section 3.2:

The validation map should be included.

Figure 8:

It would be better to show in percent.